# Colorimetric Detection of Acenaphthene and Naphthalene Using Functionalized Gold Nanoparticles

**DOI:** 10.3390/ijms24076635

**Published:** 2023-04-02

**Authors:** Kai-Jen Chuang, Meng-Ru Dong, Purnima Laishram, Gui-Bing Hong

**Affiliations:** 1School of Public Health, College of Public Health and Nutrition, Taipei Medical University, Taipei 110, Taiwan; kjc@tmu.edu.tw; 2Department of Public Health, School of Medicine, College of Medicine, Taipei Medical University, Taipei 110, Taiwan; 3Department of Chemical Engineering and Biotechnology, National Taipei University of Technology, Taipei 106, Taiwan; rube2326@gmail.com (M.-R.D.); purnimalaishram3@gmail.com (P.L.)

**Keywords:** colorimetric, gold nanoparticles, acenaphthene, naphthalene

## Abstract

Polycyclic aromatic hydrocarbons are a class of chemicals that occur naturally. They generally demonstrate a high degree of critical toxicity towards humans. Acenaphthene and naphthalene contain compounds that are commonly found in the environment as compared to other PAHs. Consequently, a reliable method of detecting PAHs is crucial for the monitoring of water quality. A colorimetric method based on sodium nitrite-functionalized gold nanoparticles was developed in this study for acenaphthene and naphthalene detection. Different functionalized parameters are determined for the optimization of assay conditions. A linear relationship was found in the analyte concentration range of 0.1–10 ppm with the limit of detection for acenaphthene and naphthalene being 0.046 ppm and 0.0015 ppm, respectively, under the optimized assay conditions. The method’s recovery rate for actual samples falls within the range of 98.4–103.0%. In selective and anti-interference tests, the presence of cations and anions has minimal impact on the detection of the analyte. The colorimetric detection method proposed in this study effectively determines the presence of the analyte in real water samples and has a high recovery rate.

## 1. Introduction

The statistics and research on cancer are quite wide-ranging, exploring the influences of factors such as genetic predisposition, lifestyle, and environmental exposures. Obesity, poor diet, lack of physical activity, and exposure to certain chemicals and pollutants have all been linked to an increased risk of cancer. Carcinogens, or substances that can cause cancer, can be found in a wide range of sources, including tobacco, alcohol, certain foods, food additives, and certain chemicals used in industry and agriculture. To protect public health, many countries have established regulations and guidelines for the detection and control of these substances. These regulations include limits on the use and release of carcinogens, as well as guidelines for monitoring exposure levels in the workplace, in food and water, and in the environment. Polycyclic aromatic hydrocarbons (PAHs) are common carcinogens that are mainly generated by the incomplete combustion of fossil fuels, biosynthetic processes of plants, pyrolysis processes in crude oil refinement, plastic production, barbecuing, thermal power generation, and so on [1,2,3]. It has been demonstrated that PAHs and their derivatives have a strong link with various cancers and have teratogenic, mutagenic, and carcinogenic risks [4,5].

As seen from the global PAH production data in the report of Li et al. [6], the final consumption quantity is recognized as a factor driving worldwide PAH-related health impacts. Huge differences in PAH production and health impacts between regions can be observed. According to these production data, the health-affected regions of China, India, and the rest of Asia produce more PAH emissions than developed regions, such as the USA, Western Europe, and East Asia [6]. In addition, PAHs can be found in a variety of environmental sources (such as vehicle exhaust, industrial emissions, and cigarette smoke), certain foods (such as grilled or smoked meats), and certain consumer products (such as creosote and tar-based roofing materials). PAHs can be harmful to human health; they can be absorbed into the body through inhalation, ingestion, or skin contact and can lead to a variety of health problems. These pollution sources are flooding into human living spaces, and even trace amounts of PAHs over a long period of exposure or ingestion can cause cancer [7]. The Commission Regulation of the European Union (EU), in amending the REACH Regulation 1907/2006 [8], listed eight PAH compounds to be controlled in consumer products. The standard method (AfPS GS 2019:01 PAK [9]) is currently used to control 15 PAHs in consumer products, and these are validated via PAH testing for GS mark certification according to the German Product Safety Law, which has been in effect since July 2020 [10]. The United States Environmental Protection Agency (US EPA) has designated 16 PAH compounds as “priority control pollutants”, and the Scientific Committee on Food (SCF) has confirmed that 15 of these PAHs are genetic carcinogens [11,12].

Chromatographic techniques such as gas chromatography/mass spectrometry (GC/MS) and high-performance liquid chromatography (HPLC) are often used to elucidate the composition of complex environmental samples [13]. However, these methods require expensive equipment, are complicated to operate, require large samples, are time-consuming and costly, involve labor-intensive sample preparation, and require a large number of organic solvents that are toxic to humans [14,15]. The limitations of these methods have led to the development of other assays, such as colorimetric sensing, which have been developed and widely used to detect various target molecules due to their simplicity, affordability, high sensitivity, and high specificity [16,17,18]. Due to the localized surface plasmon resonance (LSPR) characteristics of noble metals such as gold and silver nanoparticles, the wavelength of light absorbed by metal NPs is affected, resulting in changes in color that can be used in the colorimetric detection of pollutants [19,20]. A colorimetric method based on sodium nitrite-functionalized gold nanoparticles (AuNPs) was developed in this study for acenaphthene (Ace) and naphthalene (Nap) detection. Different functionalization parameters (e.g., amount of sodium nitrite and buffer solution, detection time) have different aggregation abilities and result in different colors, which can be used to optimize the detection conditions. Finally, the proposed colorimetric detection method was tested on real samples to evaluate its feasibility for detecting Nap and Ace.

## 2. Results and Discussion

### 2.1. Characterization of Functionalized AuNPs

AuNPs have special LSPR characteristics and can be applied to the detection of analytes. In the characterization of AuNPs and functionalized AuNPs via UV-visible absorption spectroscopy, the absorption spectra of AuNPs showed a distinctive sharp peak at 521 nm. When AuNPs were functionalized by adding NaNO_2_ and PBS buffer solution, the absorption peak remained at 521 nm. It is assignable to the surface plasmon resonance; this specifies that NaNO_2_ + AuNPs + PBS was completely diffused. When the sodium nitrite and PBS buffer solutions were added to the AuNPs solution, sodium nitrite was adsorbed onto the surface of the AuNPs. The electrostatic repulsion of nitrite and citrate on the surface of AuNPs prevented the aggregation of AuNPs, and the dispersed functionalized AuNPs solution had a strong absorbance at a wavelength of approximately 521 nm, as shown in Figure 1. When the target analyte (Ace or Nap) was added, the original absorption peak at 521 nm dropped sharply, and a new broad peak appeared at approximately 650 nm. At this time, the dispersed functionalized AuNPs solution also changed from its original dark-red color to dark blue, resulting in aggregation.

TEM can be used to observe the dispersion and aggregation of AuNPs. Figure 2a shows that the functionalized AuNPs were well dispersed and spherical with a particle size of approximately 15 nm. Adding the appropriate PBS buffer solution did not change their morphology significantly (Figure 2b). When the analyte was added to the detection solution consisting of AuNPs, NaNO_2_, and PBS buffer, the AuNPs aggregated, as shown in Figure 2c,d. The size distribution of the functionalized AuNPs (AuNPs + NaNO_2_ + PBS buffer) without the analyte is shown in Figure 2e, and the average particle size was approximately 15.78 nm. After adding the analyte to the functionalized AuNPs, the AuNPs aggregated with an average particle size of 19~20 nm, as shown in Figure 2f,g.

When the sodium citrate reduced the gold nanoparticles, the citrate anions attached to the gold nanoparticles reduced the gold ions to atoms and stabilized colloidal AuNPs formed from the cluster. The concentration relative to the gold precursor was high. In addition, NaNO_2_ is hygroscopic, so it can easily dissociate Na^+^ and NO_2_^−^ ions, a process known as the denitrification mechanism. As a consequence, the colorimetric detection of acenaphthene and naphthalene is based on the above mechanism (Figure 3). These two analytes had electrophilic substitution reactions and were more reactive. The negatively charged NO_2_^−^ ions were drawn to the selected analyte as a result of the negatively charged gold nanoparticles being drawn toward positively charged sodium ions. Therefore, the dispersed functionalized AuNPs resulted in aggregation and the solution changed from dark red to dark blue.

### 2.2. Optimization of the Amount of Sodium Nitrite

Different amounts (μ mole) of sodium nitrite solution (0.1 M) were added to the AuNPs solution for the UV-vis analysis of functionalized AuNPs. As seen in Figure 4, the absorbance of the peak at 521 nm decreased as the volume of sodium nitrite solution increased, and the redshift phenomenon became more pronounced (as shown in the inset of Figure 4). Under the premise of not affecting the stability of functionalized AuNPs, the optimal added volume of sodium nitrite is 0.5 μ mole. However, in the subsequent analyte detection test, it was found that the addition of 0.5 μ mole would lead to a long detection time, so 1.0 μ mole was selected as the optimal addition amount of sodium nitrite.

Generally, a bathochromic shift or redshift is perceived as an increasing number of aromatic rings in the molecules as determined via UV-Vis spectra. Despite that, the bathochromic effect decreased as the ring number increased. This occurrence can be interpreted as the delocalization of π electrons. The probable mechanism for a bathochromic shift was the absorption maximum being shifted towards a longer wavelength for the presence of chromophores (sodium nitrite) when sodium nitrite was added to gold nanoparticles. Figure 5 shows the probable mechanism for the bathochromic shift of NaNO_2_ + AuNPs. After calculating the bandgap energy from the UV spectra (ΔE_1_ > ΔE_2_) and comparing the maximum wavelength (λ_1_ < λ_2_), the effect was more pronounced for π-to-π* transition.

### 2.3. Optimization of the PBS Buffer Solution

Owing to the long detection time of Ace or Nap by functionalized AuNPs, we chose PBS buffer [21] instead of citrate buffer because citrate is negatively charged and provides the nanoparticles with repulsive charges. Generally, it is best not to handle the particles too much as they can aggregate easily. This is usually irreconcilable with mass spectrometric detection or may heavily interfere with ion generation. While adding PBS buffer solution can reduce the detection time, it may also impact the stability and sensitivity of functionalized AuNPs; thus, the optimal amount of added content must be determined. PBS buffer solution was added to the functionalized AuNPs solution containing 1.0 μ mole sodium nitrite in different volumes, and the experimental results are shown in Figure 6. The color of the functionalized AuNPs solution did not change when PBS buffer was added; however, the absorption peak at 521 nm decreased as the volume of added PBS buffer increased. The purpose of adding a buffer solution is to shorten the detection time while not affecting the detection performance. Therefore, the optimal addition amount of PBS buffer solution was selected as 0.3 μ mole.

### 2.4. Optimization of the Detection Time

To achieve the most time-effective detection method, the analyte (Ace and Nap) was detected with functionalized AuNPs to determine the optimal detection time. In Figure 7a, it can be seen that as the detection time for Ace increased, the absorption peak of the functionalized AuNPs at 521 nm gradually decreased, and a new absorption peak was generated. The same phenomenon was also observed in the detection of Nap, as shown in Figure 7b. Based on the detection time of the two analytes, the optimal detection time was selected as 10 min for the following investigation.

### 2.5. Sensitivity of Functionalized AuNPs

After applying the optimized conditions mentioned above, detection of the analyte was performed, and the UV-vis absorption spectra of functionalized AuNPs at different concentrations are displayed in Figure 8. It can be observed that the functionalized AuNPs solution had a redshift from the absorption peak at 521 nm, and the absorption peak at 521 nm decreased as the concentration increased. In the colorimetric detection of Ace (Figure 8a), the maximum absorption difference at 644 nm between the Ace-added and Ace-free functionalized AuNPs solutions was selected to establish the calibration curve. A linear relationship was found in the Ace concentration range of 0.1–10 ppm (the inset in Figure 8a). The linear equation is y = 1.19734 − 0.0578x, and its correlation coefficient (R^2^ value) is 0.9814. The result of the colorimetric detection of Nap is similar to that of the detection of Ace, as shown in Figure 8b. For the same reason, in the colorimetric detection of Nap, 612 nm, which has a large absorbance difference with the functionalized AuNPs solution, was selected for calibration curve analysis. The linear equation is y = 1.1488 − 0.048x (the inset in Figure 8b), and its correlation coefficient (R^2^ value) is 0.9927. According to the international standards described by the International Union of Pure and Applied Chemistry (IUPAC), the limit of detection (LOD) for Ace and Nap was determined to be 0.046 ppm and 0.0015 ppm, respectively. This colorimetric detection method was compared with the literature on Ace or Nap detection using different methods, as shown in Table 1. It can be seen from the table that the detection range of this experiment was relatively wide, simple, and can effectively detect Ace and Nap without expensive instruments, although the lowest detection limit was not the most prominent.

### 2.6. Sensitivity of Functionalized AuNPs

In addition to sensitivity, the selectivity of the detection method is also an important factor to consider. To test the selectivity of the detection method proposed in this study, analytes (Ace and Nap) and other interferents, such as different cations (Fe^2+^, Fe^3+^, Cu^2+^, Sn^4+^, Zn^2+^, K^+^, Ni^2+^) and anions (C_2_O_4_^2−^, ClO_4_^−^, CH_3_COO^−^, CO_3_^2−^, NO_3_^−^), were added to the functionalized AuNP solution, and the test was conducted under the same optimized experimental conditions. The analyte concentration was maintained at 1 ppm, while the concentration of the cations and anions was 50 ppm. The UV-vis absorbance of functionalized AuNPs in the presence of an analyte and other single ions is shown in Figure 9. The color of the functionalized AuNPs solution containing Ace (Figure 9a) and Nap (Figure 9b) changed from red to blue, while the solution containing cations and anions remained red. Although this detection method was not affected by cations and anions, it was still affected by similar structural PAH (Phe, Flu) components. Nevertheless, Ace could still be identified through UV-vis absorption spectra, as shown in Figure 10. The absorption peaks of Ace, Nap, Phe, and Flu were different; at the same concentration, Ace absorbed the least at short wavelengths (<300 nm) and the most at long wavelengths (approximately 650 nm).

### 2.7. Detection of Ace and Nap in real water

To evaluate the effectiveness of the colorimetric method, actual water samples containing Ace and Nap (from both tap and mineral sources) were analyzed. The samples were filtered before preparing solutions of Ace and Nap within the linear range of the colorimetric method. As seen in Table 2, the results show a recovery rate of 98.4–103.0% and a relative standard deviation (RSD) below 1%. These results are deemed satisfactory. The experiment demonstrated high recovery rates and low RSD values in both types of real samples, indicating that the colorimetric method proposed in this study is a promising method for detecting PAH-containing solutions.

## 3. Methods and Materials

### 3.1. Materials

Acenaphthene (Ace) (purity > 99%), phenanthrene (Phe) (purity ≥ 98%), iron(II) chloride tetrahydrate (purity > 99%), iron(III) chloride (purity > 98%), nickel(II) chloride hexahydrate (purity ≥ 97%), oxalic acid (purity ≥ 98%), sodium nitrite (purity ≥ 98.5%), sodium perchlorate (purity ≥ 99%), and tin(IV) chloride pentahydrate (purity ≥ 98%) were purchased from Acros Organics (Geel, Belgium). Acetic acid (purity ≥ 99.7%), nitric acid (purity > 68%), phosphate-buffered saline (PBS) buffer solution, and sodium carbonate (purity > 99.5%) were obtained from Fisher Chemical (Hampton, NH, USA). Copper (II) chloride (purity ≥ 98%), fluorine (Flu) (purity ≥ 98%), naphthalene (Nap) (purity ≥ 99%), and zinc chloride (purity ≥ 98%) were purchased from Alfa Aesar (Ward Hill, MA, USA). Potassium acetate (purity ≥ 99%) and tetrachloroauric acid (HAuCl_4_) (purity ≥ 99.99%) were purchased from Sigma–Aldrich (St. Louis, MO, USA). Sodium citrate (purity ≥ 99%) was obtained from Macron Fine Chemicals (Center Valley, PA, USA). All chemicals used were of analytical grade and were used without undergoing additional purification.

### 3.2. Synthesis of functionalized AuNPs

The Turkevich–Frens method [29] was used to synthesize gold nanoparticles, and the procedure was similar to that of Hong et al. [30] with some modifications. Briefly, 50 mL of tetrachloroauric acid solution (1 × 10^−3^ M) was heated in a water bath until boiling under stirring. Then, a sodium citrate solution (38.8 mM) at a volume of 5 mL was slowly added to the boiling solution, continuously heated, and stirred for approximately 10 min. The boiling solution was removed from the heating system and continuously stirred until the solution cooled to room temperature. At the same time, it was observed that the color of the solution gradually turned from light yellow to dark red, indicating that the tetrachloroauric acid solution had been reduced to a AuNP solution. Then, 600 µL of the AuNP solution was centrifuged, 500 µL of the supernatant liquid was removed, and the concentrated AuNP solution was used for functionalized AuNP preparation. Different functionalized parameters were investigated, as follows:(1)Amount of sodium nitrite: different volumes of sodium nitrite solution (0.1 M) were added to the AuNP solution (100 µL of concentrated AuNP solution + 400 µL of DI water) and mixed well to test the absorbance change with or without the analyte.(2)Amount of buffer solution: after the addition of sodium nitrite solution was determined, the addition of PBS buffer solution could be optimized.(3)Detection time: the analyte (400 µL) was added to the mixed solution of concentrated AuNP (100 µL) and the optimal contents of sodium nitrite solution and PBS buffer solution. After the mixture solution was mixed well, the absorbance was recorded using a UV-vis spectrophotometer (Thermo Scientific, Braunschweig, Germany) at different detection times.

### 3.3. Analyte Detection Based on Functionalized AuNPs

In a typical experiment, 400 µL of different concentrations (0.1–10 ppm) of analyte (Nap and Ace) solutions were added to 100 µL of concentrated AuNP solutions in centrifuge tubes (1.5 mL) and mixed well. Then, a sodium nitrite solution (1.0 μ mole solute) and PBS buffer solution (0.3 μ mole solute) were added. After the solution was evenly mixed for 10 min, the absorbance of the solution was measured with a UV-vis spectrophotometer. A variety of common substances that may exist in samples were tested for the selectivity and anti-interference ability of the functionalized AuNPs. The concentration of these chemical substances was set at 50 mg/L, and the experimental procedures were identical to those used in the detection method for the analyte.

## 4. Conclusions

This study used a colorimetric method to detect Ace and Nap based on a functionalized AuNPs solution. When sodium nitrite and PBS buffer solution were added to the AuNPs solution, sodium nitrite adsorbed onto the surface of the AuNPs and then formed covalent bonds with the analyte, resulting in aggregation and a change in color from dark-red to dark blue. A linear relationship was found in the analyte concentration range of 0.1–10 ppm with the LOD for Ace and Nap being 0.046 ppm and 0.0015 ppm, respectively. The recovery rate for real samples was in the range of 98.4–103.0%. In selective and anti-interference experiments, cations and anions had almost no effect on the detection of the analyte. Despite interference from similar structural PAHs components, the proposed colorimetric detection method still has a certain degree of reliability in their detection. The method can be effectively applied to detect an analyte in real water samples and has a high recovery rate. Additionally, polycyclic aromatic hydrocarbons generally exhibit a high degree of critical toxicity towards humans. As compared to other PAHs, acenaphthene and naphthalene contain compounds that are commonly found in the environment. So, this colorimetric detection method helps us to detect unwanted Nap and Ace in an effective, simple, highly sensitive, environmentally friendly, and easy-to- prepare manner.

## Figures and Tables

**Figure 1 ijms-24-06635-f001:**
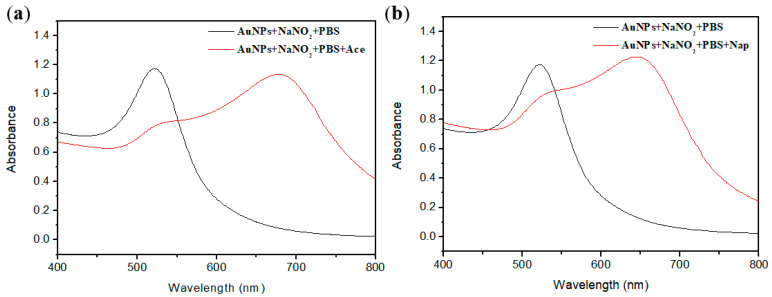
UV-vis absorption spectra before and after adding the target analyte: (**a**) Ace and (**b**) Nap.

**Figure 2 ijms-24-06635-f002:**
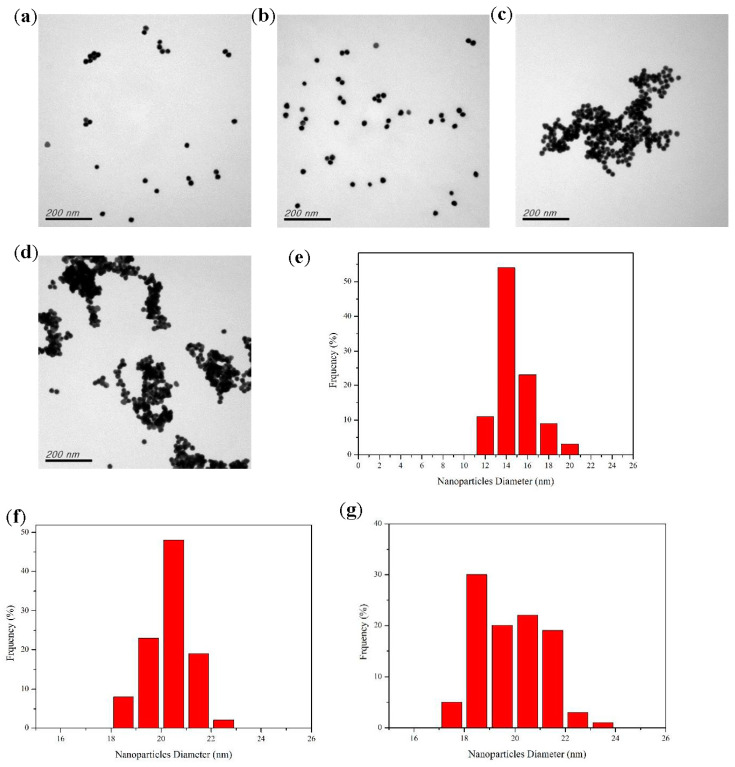
TEM images of (**a**) AuNPs + NaNO_2_, (**b**) AuNPs + NaNO_2_ + PBS buffer, (**c**) AuNPs + NaNO_2_ + PBS buffer + Ace, (**d**) AuNPs + NaNO_2_ + PBS buffer + Nap, and the size distribution of (**e**) AuNPs + NaNO_2_ + PBS buffer, (**f**) AuNPs + NaNO_2_ + PBS buffer + Ace, and (**g**) AuNPs + NaNO_2_ + PBS buffer + Nap.

**Figure 3 ijms-24-06635-f003:**
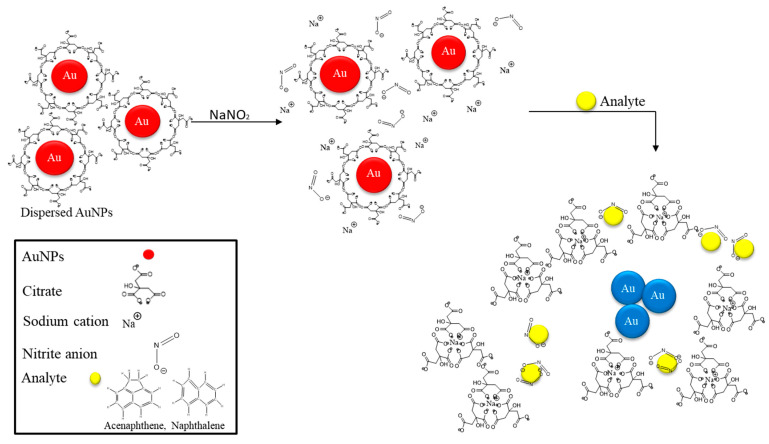
Graphic sketch of Ace and Nap detection using functionalized AuNPs.

**Figure 4 ijms-24-06635-f004:**
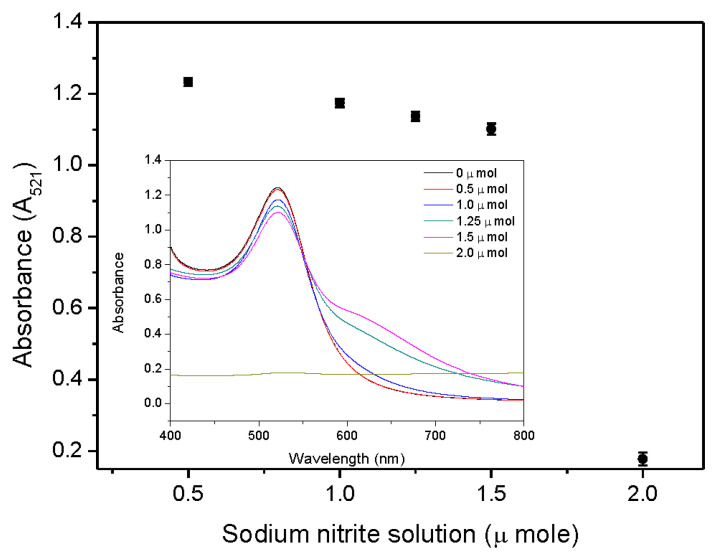
Absorbance at 521 nm of the blank test (without the existence of analyte) with different volumes of sodium nitrite solution. The inset shows the UV-vis spectra of functionalized AuNPs.

**Figure 5 ijms-24-06635-f005:**
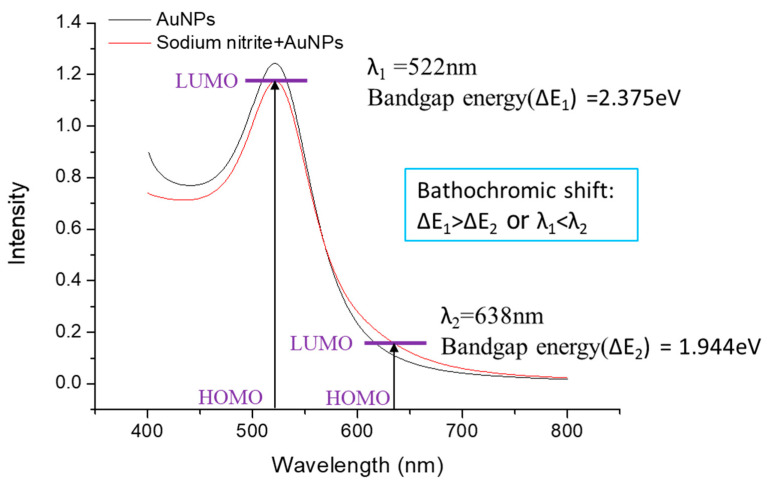
Probable mechanism for the bathochromic shift of NaNO_2_ + AuNPs.

**Figure 6 ijms-24-06635-f006:**
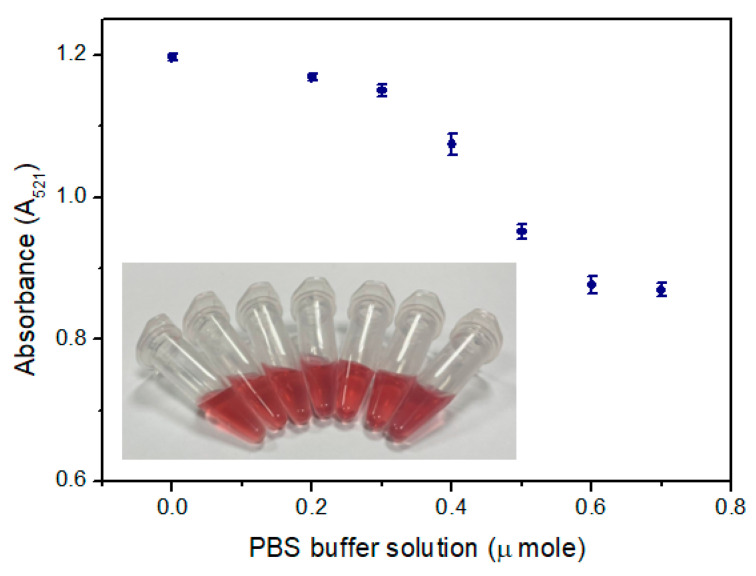
Absorbance at 521 nm of the blank test (without the existence of analyte) with different volumes of PBS buffer solution.

**Figure 7 ijms-24-06635-f007:**
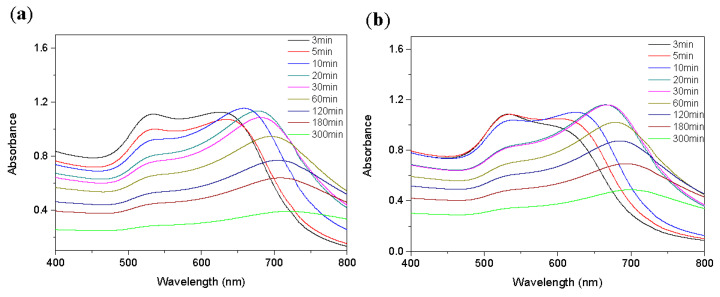
Detection times for (**a**) Ace and (**b**) Nap.

**Figure 8 ijms-24-06635-f008:**
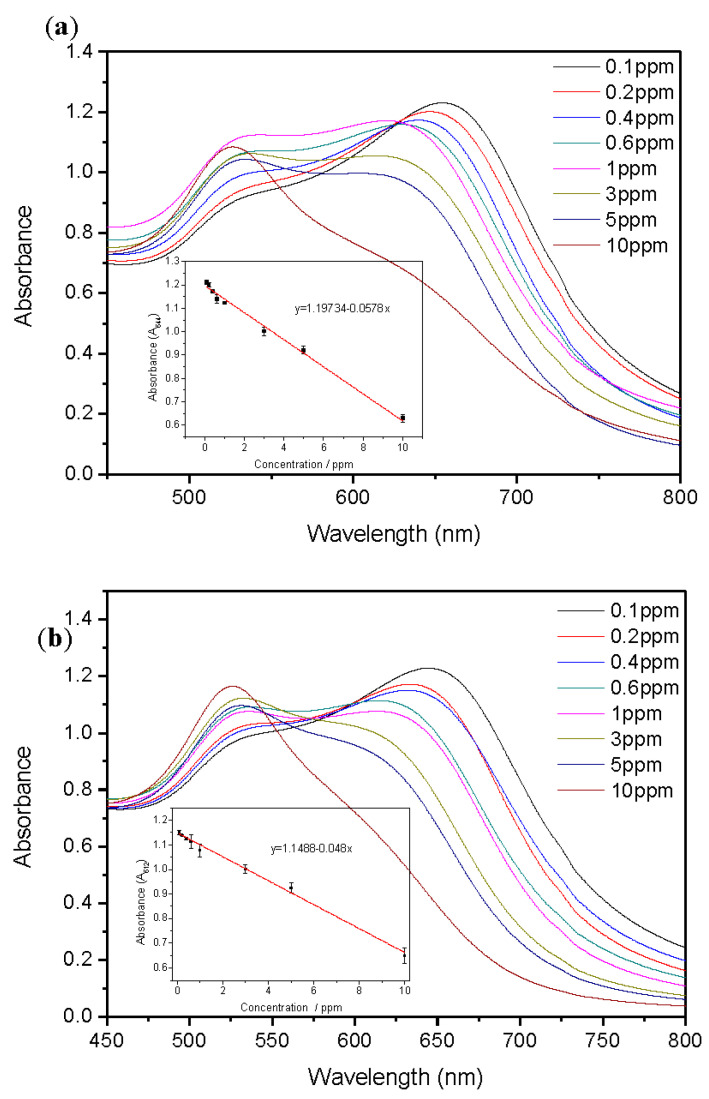
UV-vis absorption spectra of functionalized AuNPs under different concentrations of (**a**) Ace and (**b**) Nap.

**Figure 9 ijms-24-06635-f009:**
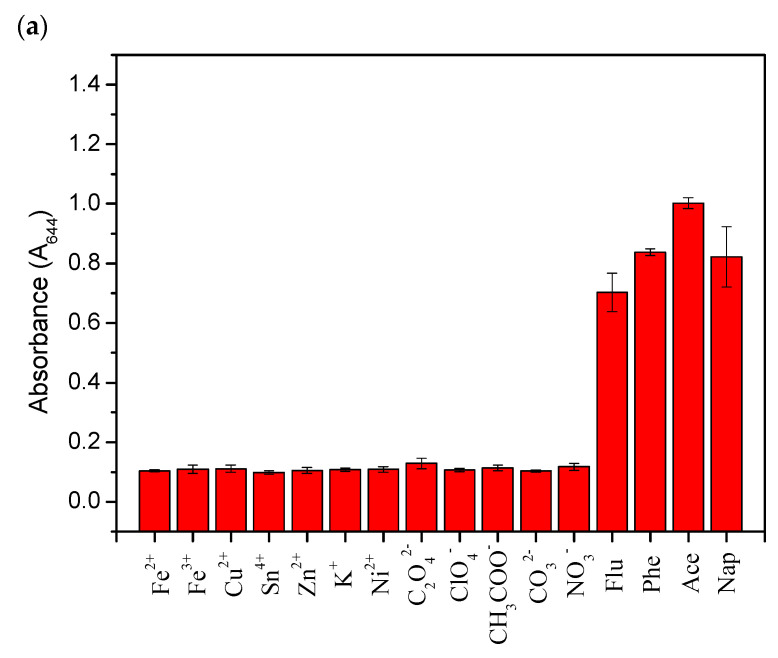
UV-vis absorbance of functionalized AuNPs in (**a**) the presence of Ace and other single ions and (**b**) the presence of Nap and other single ions.

**Figure 10 ijms-24-06635-f010:**
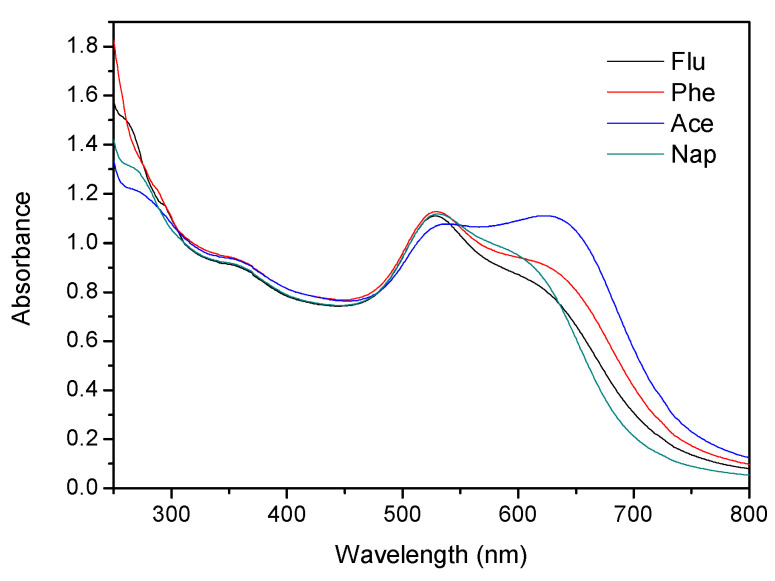
UV-vis absorbance of functionalized AuNPs in the presence of Flu, Phe, Ace, and Nap.

**Table 1 ijms-24-06635-t001:** Comparison of LOD for analytes with literature values.

Analytes	Methods	Linear Range (ppb)	LOD (ppb)	Detection Time	Detection Cost	Ref.
Ace	MS	2.05–100	0.61	>7 min	High	[22]
	GC	10–1000	0.2	<10 min	High	[23]
	SPE-HPLC	0.5–10	0.008	<17 min	High	[24]
	HPLC	1000–7000	880	15 min	High	[25]
	Colorimetric	100–10,000	46	Quick	Low	This study
Nap	Bioluminescence	50–500	-	70 min	Medium	[26]
	SERS	1.28–12.8	0.871	<15 h	High	[27]
	SPE-HPLC	0.5–10	0.04	<17 min	High	[24]
	HPLC	1000–7000	1320	15 min	High	[25]
	Immuno-PCR	1–10,000	-	Quick	Low	[28]
	Colorimetric	100–10,000	1.5	Quick	Low	This study

**Table 2 ijms-24-06635-t002:** Recovery of analytes in water samples.

Samples	Analytes	Added (ppm)	Found (ppm)	Recovery (%)	RSD (%)
Tap water	Ace	3	2.97 ± 0.14	98.9	±0.81
Mineral water		3	2.95 ± 0.18	98.4	±0.98
Tap water	Nap	3	3.09 ± 0.19	103.0	±0.82
Mineral water		3	3.01 ± 0.15	100.4	±0.69

## Data Availability

Not applicable.

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
