# Peer review of "Colorimetric Detection of Acenaphthene and Naphthalene Using Functionalized Gold Nanoparticles"

_ijms, 2023, doi:10.3390/ijms24076635_

Round 1

Reviewer 1 Report

In this manuscript, Chuang and co-workers established the colorimetric determination of Nap and Ace by sodium nitrite functionalized Au NPs, which can be applied to the determination of some analytes in water with high recovery rates. However, the study lacks obvious data to interpretate the rusults. Therefore, major revisions should be made before this article can be accepted for publication.

Comments:

1. In Part 2.3, authors believe the optimal addition amount of PBS buffer is 3 μl. More detailed interpretation or experiments should be given to verify this point.

2. Authors select 521 nm to analysis a series of experiments. I thick more evidences should be given about the rationality of this choice.

3. Please supplement the origin of the linear relation in Figure 6.

4. Comparison of more specific items should be given in Table 1.

5. Please provide size distribution of different NPs in Figure 2.

6. The conclusion part is only summarized based on the experimental results, and the contribution and significance about this work are insufficient. Please summarize it further.

Author Response

Point 1: In Part 2.3, authors believe the optimal addition amount of PBS buffer is 3 μl. More detailed interpretation or experiments should be given to verify this point.

Response 1: The interpretation was added in Part 3.3 “Optimization of the PBS buffer solution” of the revised manuscript.

Point 2: Authors select 521 nm to analysis a series of experiments. I thick more evidences should be given about the rationality of this choice.

Response 2: The characterization of AuNPs and functionalized AuNPs by UV-Visible absorption spectroscopy, the absorption spectra of AuNPs was shown a distinctive sharp peak at 521nm. When AuNPs were functionalized by adding NaNO2 and PBS buffer solution, the absorption peak was shown still at 521 nm. It is assignable to the surface plasmon resonance, this specifies that NaNO2+AuNPs+PBS were completely diffused. As suggested by the reviewer, we have added more evidence in the revised manuscript. Please see Part 3.1 in the revised manuscript.

Point 3: Please supplement the origin of the linear relation in Figure 6.

Response 3: Fig. 9 (a) and (b) shows the origin of the linear relation. The origin of the linear relationship will be the functionalized AuNPs without an analyte.

Point 4: Comparison of more specific items should be given in Table 1.

Response 4: As suggested by the Reviewer, we have added specific items in Table 1 of the revised manuscript.

Point 5: Please provide size distribution of different NPs in Figure 2.

Response 5: Very appreciate the comment of the Reviewer. The size distribution of different AuNPs was added in Fig. 3 of the revised manuscript.

Point 6: The conclusion part is only summarized based on the experimental results, and the contribution and significance about this work are insufficient. Please summarize it further.

Response 6: As the reviewer suggested, we have put more in the conclusion part.

Reviewer 2 Report

The communication "Colorimetric detection of acenaphthene and naphthalene using functionalized gold nanoparticles" describes a simple colorimetric method for the determination of polycyclic hydrocarbons using functionalized gold nanoparticles. The work is at a good level, but there are some comments about it.

Major point:

The selectivity of the proposed method for the determination of acenaphthene and naphthalene concerning other aromatic compounds, such as benzene, anthracene, pyrene, and others, is not shown in the work. Without it, it is impossible to speak about any selectivity of the offered method of definition of compounds. In addition, the authors refer to their article on the determination of pyrene and phenanthrene using gold nanoparticles functionalized with ammonium nitrate (https://www.sciencedirect.com/science/article/pii/S1386142521012786?via%3Dihub). However, they do not show the selectivity of recognition of acenaphthene  and naphthalene compared to pyrene and phenanthrene in this work. Also, the effect of acenaphthene and naphthalene on each other in the combined presence was not demonstrated (Fig. 7).

Minor points:

1)            Figure 2 shows NH4NO3 in the caption, but the text when describing Figure 2 says sodium nitrite. What salt was added to produce the TEM images?

2)            Give a probable mechanism for the bathochromic shift when sodium nitrite is added to gold nanoparticles.

3)            It is not very clear why there is one detection limit for two calibration curves with different slopes. An explanation is required.

Author Response

Point 1: The selectivity of the proposed method for the determination of acenaphthene and naphthalene concerning other aromatic compounds, such as benzene, anthracene, pyrene, and others, is not shown in the work. Without it, it is impossible to speak about any selectivity of the offered method of definition of compounds. In addition, the authors refer to their article on the determination of pyrene and phenanthrene using gold nanoparticles functionalized with ammonium nitrate (https://www.sciencedirect.com/science/article/pii/S1386142521012786?via%3Dihub). However, they do not show the selectivity of recognition of acenaphthene and naphthalene compared to pyrene and phenanthrene in this work. Also, the effect of acenaphthene and naphthalene on each other in the combined presence was not demonstrated (Fig. 7).

Response 1: Very appreciate the comment of the Reviewer. The selectivity of recognition of acenaphthene and naphthalene is very important in this work. Therefore, the selectivity test results were compared to phenanthrene (Phe) and fluorine (Flu), as shown in Figs. 10-11. Although this detection method is not affected by cations and anions, it is still disturbed by similar structural PAH (Phe, Flu) components. Nevertheless, Ace can still be identified through UV‒vis absorption spectra, as shown in Fig. 11. The absorption peaks of Ace, Nap, Phe, and Flu are different; at the same concentration, Ace absorbs the least at short wavelengths (<300 nm) and the most at long wavelengths (approximately 650 nm).

Point 2: Figure 2 shows NH4NO3 in the caption, but the text when describing Figure 2 says sodium nitrite. What salt was added to produce the TEM images?

Response 2: The TEM images were produced by adding sodium nitrite. The mistake was corrected in the revised manuscript.

Point 3: Give a probable mechanism for the bathochromic shift when sodium nitrite is added to gold nanoparticles.

Response 3: Generally, bathochromic shift or redshift is perceived as the number of aromatic rings increases in the molecules by the detection of UV spectra. Despite that, the bathochromic effect decreases as the ring number increases. This occurrence can be interpreted by the delocalization of π electrons. The probable mechanism for a bathochromic shift was the absorption maximum being shifted towards a longer wavelength for the presence of chromophores (sodium nitrite) when sodium nitrite is added to gold nanoparticles. Fig. 6 shows the probable mechanism for the bathochromic shift of NaNO2 + AuNPs. After calculating the bandgap energy from the UV spectra (ΔE1>ΔE2) and comparing the maximum wavelength (λ12), the effect is more pronounced for π to π* transition.

Point 4: It is not very clear why there is one detection limit for two calibration curves with different slopes. An explanation is required.

Response 4: Very appreciate the comment of the Reviewer. The limit of detection (LOD) for Ace and Nap are 0.046 ppm and 0.0015 ppm, respectively, under the optimized assay conditions. The mistake was corrected in the revised manuscript.

Reviewer 3 Report

The manuscript (2304301) entitled ‘‘Colorimetric detection of acenaphthene and naphthalene using functionalized gold nanoparticles’’ can be published in the International Journal of Molecular Sciences after minor revision.  

My suggestions:

1-      You should draw the open structure of acenaphthene and naphthalene. Let's understand the adsorption spectrum easily and why these analytes can be removed with AuNPs?

2-      Page 3, line 95, NaNO2 shoud be corrected as NaNO2

3-      Page 7, line 153 and 157, R2 should be corrected as R2

4-      The indices in the chemical compounds in Figure 2 should be corrected as subscripts. For example, AuNPs+NH4NO3 should be corrected as AuNPs+NH4NO3,….

5-      There is no comparison chart for the removal of ions in Figure 7. I think this is a shortcoming. This activity of AuNPs should be compared with other adsorbents. Let me suggest a ref at this stage (please look at the Tables in this suggested ref. 1-Journal of Polymers and the Environment, 29(11), 3477-3496.

Author Response

Point 1: You should draw the open structure of acenaphthene and naphthalene. Let's understand the adsorption spectrum easily and why these analytes can be removed with AuNPs?

Response 1: When the sodium citrate reduced the gold nanoparticles, the citrate anions attached to the gold nanoparticles reduce the gold ions to atoms and stabilize colloidal AuNPs formed from the cluster. The concentration relative to gold precursor is high. In addition, NaNO2 is hygroscopic so that can easily dissociate Na+ and NO2ions which are the process of denitrification mechanism. As a consequence, the colorimetric detection of acenaphthene and naphthalene is based on the above mechanism (Fig. 4). These two analytes were electrophilic substitution reactions that occurred and were more reactive. The negatively charged NO2ions captivated with the selected analyte as a result of the negatively charged gold nanoparticles being captivated with positively charged sodium ions.

Point 2: Page 3, line 95, NaNO2 should be corrected as NaNO2

Response 2: As suggested by the Reviewer, we have corrected this mistake in the revised manuscript.

Point 3: Page 7, line 153 and 157, R2 should be corrected as R2

Response 3: As suggested by the Reviewer, we have corrected it in the revised manuscript.

Point 4: The indices in the chemical compounds in Figure 2 should be corrected as subscripts. For example, AuNPs+NH4NO3 should be corrected as AuNPs+NH4NO3,….

Response 4: As suggested by the Reviewer, we have corrected the format in the revised manuscript.

Point 5: There is no comparison chart for the removal of ions in Figure 7. I think this is a shortcoming. This activity of AuNPs should be compared with other adsorbents. Let me suggest a ref at this stage (please look at the Tables in this suggested ref. 1-Journal of Polymers and the Environment, 29(11), 3477-3496.

Response 5: Very appreciate the comment of the Reviewer. The selectivity of recognition of acenaphthene and naphthalene is very important in this work. Therefore, the selectivity test results were compared to phenanthrene (Phe) and fluorine (Flu), as shown in Figs. 10-11. Although this detection method is not affected by cations and anions, it is still disturbed by similar structural PAH (Phe, Flu) components. Nevertheless, Ace can still be identified through UV‒vis absorption spectra, as shown in Fig. 11. The absorption peaks of Ace, Nap, Phe, and Flu are different; at the same concentration, Ace absorbs the least at short wavelengths (<300 nm) and the most at long wavelengths (approximately 650 nm).

Reviewer 4 Report

The manuscript entitled "Colorimetric detection of acenaphthene and naphthalene using functionalized gold nanoparticles” is an interesting piece of study that is focused on the gold nanoparticle mediated detection of two emerging pollutants. However, the quality of the manuscript can be further strengthened by addressing the following queries:

1.      In the abstract section, lines 14-15 does not make sense. Kindly revise the statement.

2.      In the abstract section, authors should remove the abbreviation as it is not required.

3.      In the introduction section, the author should include the global production data of the PAHs that make the present study interesting.

4.      in the 2.3 section, why citrate buffer is not used. Gold nanoparticles are predominately buffer in gold nanoparticles.

5.      In Figures 3 and 4, the x-axis should not be in the microliter. However, it should be in ppm/mM for better presentation of the data.

6.      in the 2.3 section, why citrate buffer is not used. Gold nanoparticles are predominately buffer in gold nanoparticles.

7.      In figure 7 a and b, authors should discuss the reason why the trends for the detection of Ace and Nap appeared the same. Secondly, the concentration of the x-axis is 1 ppm.

8.      References are not uniform. Abbreviation of journal name should be corrected.

Author Response

Point 1: In the abstract section, lines 14-15 does not make sense. Kindly revise the statement.

Response 1: Very appreciate the comment of the Reviewer. The abstract was revised in the revised manuscript.

Point 2: In the abstract section, authors should remove the abbreviation as it is not required.

Response 2: The abbreviation in the abstract was removed in the revised manuscript.

Point 3: In the introduction section, the author should include the global production data of the PAHs that make the present study interesting.

Response 3: From the global PAH production data in Fig. 1, the final consumption quantity was recognized as the factor driving worldwide PAH-related health impacts. The huge differences in PAHs production and health impacts between regions can be observed. From the interpretation of production, PAH emission and health-affected regions including China, India and the rest of Asia were more huge than the developed regions such as the USA, Western Europe, and East Asia [6]. As suggested by the Reviewer, the above sentences were added to the revised manuscript.

Point 4: In the 2.3 section, why citrate buffer is not used. Gold nanoparticles are predominately buffer in gold nanoparticles.

Response 4: Owing to the long detection time of Ace or Nap by functionalized AuNPs, we choose PBS buffer [23] instead of citrate buffer because citrate is negatively charged and provides the nanoparticles with repulsive charges. Generally, it is best not to do too much handling of the particles as they can aggregate easily. It is usually irreconcilable with mass spectrometric detection or may heavily interfere with ion generation. As suggested by the Reviewer, the above sentences were added to the revised manuscript.

Point 5: In Figures 3 and 4, the x-axis should not be in the microliter. However, it should be in ppm/mM for better presentation of the data.

Response 5: These figures (Fig. 5 and Fig. 7) were revised in the manuscript.

Point 6: In the 2.3 section, why citrate buffer is not used. Gold nanoparticles are predominately buffer in gold nanoparticles.

Response 6: Owing to the long detection time of Ace or Nap by functionalized AuNPs, we choose PBS buffer [23] instead of citrate buffer because citrate is negatively charged and provides the nanoparticles with repulsive charges. Generally, it is best not to do too much handling of the particles as they can aggregate easily. It is usually irreconcilable with mass spectrometric detection or may heavily interfere with ion generation. As suggested by the Reviewer, the above sentences were added to the revised manuscript.

Point 7: In figure 7 a and b, authors should discuss the reason why the trends for the detection of Ace and Nap appeared the same. Secondly, the concentration of the x-axis is 1 ppm.

Response 7: The figure was revised in the manuscript.

Point 8: References are not uniform. Abbreviation of journal name should be corrected.

Response 8: The references were corrected in the manuscript.

Round 2

Reviewer 1 Report

 Accept .

Reviewer 2 Report

Minor point:

Page 4, line 140; Page 5, lines 180, 184; Page 10, lines 279-280, and so on. Ions must have charges and a number of atoms in the superscript or subscript. 

The authors have fully responded to my comments and I think the manuscript can be accepted in its current form.